**Cite this article:** von Rueden CR, Redhead D, O'Gorman R, Kaplan H, Gurven M. 2019 The dynamics of men's cooperation and social status in a small-scale society. *Proc. R. Soc. B* **286**: 20191367.

behaviour, cognition, evolution

cooperation, social status, hierarchy, social networks, egalitarianism

**Authors for correspondence:**
Christopher R. von Rueden
e-mail: cvonrued@richmond.edu
Daniel Redhead
e-mail: daniel_redhead@eva.mpg.de

†Joint first author.

# The dynamics of men's cooperation and social status in a small-scale society

Christopher R. von Rueden[1,†], Daniel Redhead[2,3,†], Rick O'Gorman[3], Hillard Kaplan[4] and Michael Gurven[5]

[1]Jepson School of Leadership Studies, University of Richmond, 221 Richmond Way, Richmond, VA 23173, USA
[2]Department of Human Behavior, Ecology and Culture, Max Planck Institute for Evolutionary Anthropology, Leipzig, Germany
[3]Department of Psychology, University of Essex, Wivenhoe Park, Colchester CO4 3SQ, UK
[4]Economic Science Institute, Chapman University, One University Drive, Orange, CA 92866, USA
[5]Department of Anthropology, University of California, Santa Barbara, CA 93106, USA

CRvR, 0000-0002-3225-5791; DR, 0000-0002-2809-8121; MG, 0000-0002-5661-527X

We propose that networks of cooperation and allocation of social status co-emerge in human groups. We substantiate this hypothesis with one of the first longitudinal studies of cooperation in a preindustrial society, spanning 8 years. Using longitudinal social network analysis of cooperation among men, we find large effects of kinship, reciprocity and transitivity in the nomination of cooperation partners over time. Independent of these effects, we show that (i) higher-status individuals gain more cooperation partners, and (ii) individuals gain status by cooperating with individuals of higher status than themselves. We posit that human hierarchies are more egalitarian relative to other primates species, owing in part to greater interdependence between cooperation and status hierarchy.

## 1. Introduction

Humans give more generously when observed by others [1,2]. Evolutionary explanations for the appeal of such conspicuous generosity include costly signalling [3], indirect reciprocity [4], reputation-based partner choice [5] and service-for-prestige [6,7]. According to these models, a reputation for generosity results in group-wide favouritism in subsequent exchange, advantageous alliances, political influence or mating opportunities. In more general terms, building a reputation for greater ability and willingness to benefit others (i.e. prestige) grants greater access to contested material and social resources (i.e. social status). Prestige has greater social currency in humans relative to other apes [8,9], for whom an ability and willingness to inflict costs on others (i.e. dominance) tends to be a stronger determinant of status [10]. Humans evolved greater interdependence, relying on each other for learning skills, producing food, engaging in mutual defence and raising offspring [11]. Individuals who can offer unique services in these contexts may receive deference or are preferred cooperation partners and mates [12]. Humans also evolved the cognitive abilities to create weapons and form large levelling coalitions that could check would-be dominants [13].

Field studies provide suggestive evidence that status accrues to those who build prestige via cooperation: individuals who receive more social support or wield more informal political influence are often generous food producers [14–20], sharers of valued information [21,22] or generous contributors to collective action [23–26]. However, these field studies are cross-sectional, so they lack tests of dynamics and their interpretation is subject to potential endogeneity or reverse causality issues [27]. Indeed, cooperation is as likely to be a consequence as to be a cause of status differences, for at least four reasons. First, greater access to contested resources enhances individuals' ability to build prestige and increase their social following via cooperation. This is especially likely given the 'Matthew effect', whereby high-status individuals are evaluated more favourably or

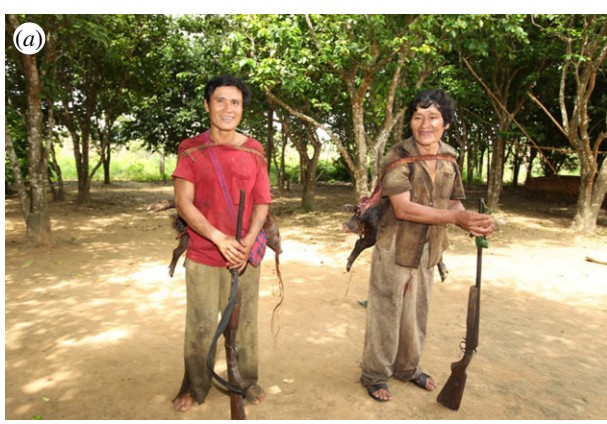
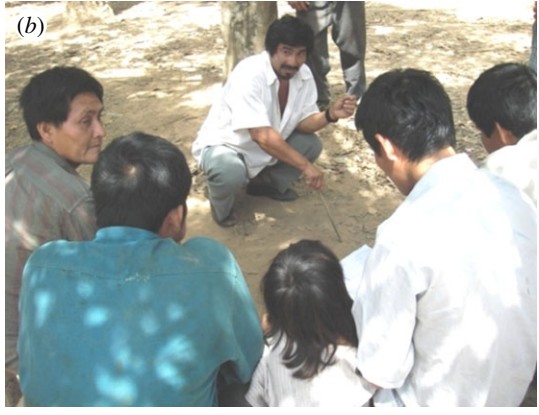

**Figure 1.** (*a*) Two Tsimane men returning from a hunt; (*b*) Tsimane man helping resolve a dispute over land, which is illustrative of the largely informal way in which political influence operates in this society. Photo credits: Chris von Rueden. (Online version in colour.)

experience other advantages owing to their status [28]. Status can be self-reinforcing. Second, the status may motivate continued generosity, in order to signal that one's status is justified or one's leadership is trustworthy [29]. Such generosity on the part of high-status individuals may become normative, i.e. 'noblesse oblige' [30]. Third, cooperation by high-status individuals may motivate cooperation by other group members, owing to prestige-biased imitation [31,32]. Fourth, by forming cooperative partnerships with high-status individuals, group members elevate their own status by gaining access to high-status individuals' information [10], resources or coalitions.

Thus, cooperation between individuals may be driven by opportunities to acquire status, and the distribution of status within a group can affect how and with whom individuals choose to cooperate. We contend that the emergence of cooperation in human groups is influenced by status hierarchy, just as the emergence of the human status hierarchy is shaped by opportunities for mutually beneficial cooperation. We further posit that human hierarchies (at least in many small-scale societies) may be more egalitarian relative to the hierarchies of other primate species [12,13], owing in part to greater dependence of status on cooperation. There is a market-like competition between higher-status individuals, and also between those low in status, with high-status individuals aiming to increase or maintain their followership via cooperation and those low in status seeking high-status cooperation partners to improve their own status. By contrast, high-ranking non-human primates are less likely to cooperate with lower-ranking individuals (in terms of grooming or food sharing), particularly in species where status is not heavily dependent on coalition formation [33,34].

We derive two predictions from the foregoing theory, which we test in a small-scale human society using longitudinal social network analysis: (i) higher-status individuals will be more frequently nominated as a cooperation partner, and (ii) over time, individuals increase their social status by cooperating with higher-status individuals. The first prediction reflects a process of network selection (tie formation) while the second prediction reflects a process of network influence (ties affect individuals' attributes). Both our predictions depart from models of the evolution of cooperation that rely on network selection but between similar behavioural types [35,36]. Instead, we posit tie formation predominantly between individuals of initially different statuses and, in turn, similarity of connected individuals resulting more from network influence than from network selection.

Tests of these predictions use longitudinal data spanning 8 years, collected among Tsimane forager-horticulturalists in Amazonian Bolivia. The Tsimane are relatively egalitarian, but men ostensibly wield more informal political influence than women, in part owing to a division of labour that typically constrains women's networking beyond the extended household [37]. For this study, we focused our attention on men. We collected three waves of panel data on all men in one village (aged 21 years and older, ave. $n = 80$). Wave 1 of the data was collected in 2009, wave 2 in 2014 and wave 3 in 2017. At each wave, men listed other adult men who shared food with them or assisted them in hunting, fishing or horticultural labour over the previous six months (figure 1*a*). Roughly two weeks later, a sample of these men photo-ranked all other adult men on two measures of status: who is respected and who has influence during village meetings (figure 1*b*). We combined these photo-ranked status measures into a single variable (status), as suggested by a maximum-likelihood factor analysis (electronic supplementary material). Peer-ratings can be an efficient and accurate method for producing quantitative data from local knowledge [38,39], especially for a public, positional good-like status. Furthermore, studies of the Tsimane and other small-scale societies find that peer-rated status correlates strongly with observational measures of status [37,40].

Our analytical approach is stochastic actor-oriented modelling (SAOMs) [41]. SAOMs estimate the node-level (actor), dyad-level (partner) and network-level (group) mechanisms that are associated with a change in network ties and in dependent attributes (behaviours) of actors (see the electronic supplementary material, table S3 for a description of specified effects in our model). To test our predictions, status is analysed in association with change in network ties and concurrently assessed as a dependent 'behaviour'. Our analysis was necessarily restricted to those men present in at least two time waves (ave. $n = 68$). As a reliability check, we include men's status in a cross-sectional analysis of the cooperation network in another Tsimane community ($n = 89$).

## 2. Results

Note that all in-degree parameters in our model capture the tendency for an individual to be nominated as a cooperation partner. Out-degree parameters indicate whether an individual is more likely to nominate others as cooperating with

**Table 1.** Estimated effects of network structure and covariates on cooperation network dynamics and status dynamics, from a network-behaviour coevolutionary stochastic actor-oriented model (SAOM). $n = 60$ at $T_1$ (2009), 74 at $T_2$ (2014), 70 at $T_3$ (2017).

| parameter | $\beta$ | s.e | $p$ | OR (CI) |
|---|---|---|---|---|
| *cooperation network dynamics* | | | | |
| cooperation rate (period 1) | 35.12 | 8.56 | <0.001 | — |
| cooperation rate (period 2) | 11.51 | 1.28 | <0.001 | — |
| out-degree (density) | 0.58 | 0.46 | 0.211 | 1.78 (0.72, 4.39) |
| reciprocity | 1.52 | 0.12 | <0.001 | 4.59 (3.60, 5.85) |
| tendency towards transitivity[a] | 1.43 | 0.10 | <0.001 | 4.17 (3.46, 5.03) |
| in-degree popularity (sqrt) | −0.54 | 0.11 | <0.001 | 0.58 (0.47, 0.72) |
| out-degree activity (sqrt) | −0.57 | 0.12 | <0.001 | 0.57 (0.45, 0.72) |
| main effect of kinship | 0.59 | 0.10 | <0.001 | 1.81 (1.48, 2.20) |
| status in-degree | 0.06 | 0.03 | 0.038 | 1.06 (1.00, 1.13) |
| status out-degree | 0.11 | 0.05 | 0.013 | 1.12 (1.02, 1.22) |
| status similarity | 0.31 | 0.22 | 0.164 | 1.36 (0.88, 2.08) |
| strength and size in-degree | 0.18 | 0.06 | 0.003 | 1.20 (1.06, 1.36) |
| strength and size out-degree | −0.04 | 0.08 | 0.611 | 0.96 (0.82, 1.12) |
| strength and size similarity | 0.35 | 0.23 | 0.131 | 1.42 (0.90, 2.22) |
| income in-degree | −0.02 | 0.02 | 0.362 | 0.98 (0.95, 1.02) |
| income out-degree | −0.01 | 0.02 | 0.812 | 0.99 (0.95, 1.04) |
| income similarity | 0.25 | 0.16 | 0.112 | 1.28 (0.94, 1.74) |
| log age in-degree | 0.02 | 0.33 | 0.961 | 1.02 (0.53, 1.95) |
| log age out-degree | −3.81 | 0.67 | <0.001 | 0.02 (0.01, 0.08) |
| log age similarity | −0.53 | 0.24 | 0.029 | 0.59 (0.36, 0.95) |
| *status dynamics* | | | | |
| status rate (period 1) | 7.60 | 1.96 | <0.001 | — |
| status rate (period 2) | 6.62 | 1.76 | <0.001 | — |
| status linear shape | −0.01 | 0.07 | 0.9 | 0.99 (0.86, 1.14) |
| status quadratic shape | −0.03 | 0.02 | 0.114 | 0.97 (0.93, 1.01) |
| status average alter | 0.19 | 0.09 | 0.037 | 1.20 (1.01, 1.43) |
| strength and size | 0.17 | 0.08 | 0.037 | 1.19 (1.01, 1.40) |
| income | 0.06 | 0.02 | 0.008 | 1.06 (1.02, 1.11) |
| log age | 0.90 | 0.46 | 0.048 | 2.46 (1.01, 5.99) |

[a]Tendency towards transitivity was measured using the geometrically weighted shared edgewise partners effect (GWESP), where $\alpha = 0.69$. For further elaboration of the effect, see the electronic supplementary material.

him. This departs from the usual terms used within the cooperation literature, with in-degree and out-degree being applied to the direction of the flow of aid, not the direction of actor nominations. However, our approach is synonymous with other studies employing SAOMs, where measurement of cooperation captures an actor's perception of their cooperative relationships and the underlying events that comprise such relationships (e.g. the actual food transfers/aid in production) are latent/unobserved. See table 1 for the effects of all parameters in our SAOM.

## (a) Within-network effects on cooperation dynamics
There was a substantial rate of change in cooperation ties throughout the study. Changes peaked between waves 1 and 2 (see rate parameters in table 1). Over time, nominations of co-operation were likely to be reciprocated (odds ratio (OR) = 4.59,

confidence interval (CI) = (3.60, 5.85)). Our results further indicate that individuals were more likely to form transitive groups of cooperators (OR = 4.17, CI = (3.46, 5.03)). This suggests that if individual *i* nominated individual *j* and individual *j* nominates individual *h*, then the odds of individual *i* subsequently nominating individual *h* as a cooperator increased. In-degree popularity (OR = 0.58, CI = (0.47, 0.72)) and out-degree activity (OR = 0.57, CI = (0.45, 0.72)) decreased the odds of ties forming. In other words, those receiving many nominations as a cooperation partner, and individuals nominating many others as cooperation partners, were 42% and 43% less likely to receive or provide nominations (respectively) in a later period.

## (b) Actor covariate effects on cooperation dynamics
Being close kin increased the odds of cooperation partnerships forming (OR = 1.81, CI = (1.48, 2.20)). Being physically stronger

and larger increased the odds of being nominated as a cooperation partner (OR = 1.20, CI = (1.06, 1.36)), but there was no substantial association between physical strength and size and the odds of out-degree nominations, nor was there a meaningful tendency for cooperation partnerships to form between individuals similar in physical strength and size. There was no considerable association between income and the odds of sending or receiving cooperation nominations, nor did individuals of similar income tend to prefer each other as cooperation partners. Results further suggest that log age had no considerable association with the odds of receiving nominations as a cooperation partner, but being older tended to decrease the odds of naming others as cooperation partners (OR = 0.02, CI = (0.01, 0.08)). Cooperation partnerships were 41% less likely to form between individuals of a similar age (OR = 0.59, CI = (0.36, 0.95)). In other words, the results suggest a tendency for individuals to form cooperation partnerships with peers dissimilar to themselves in age.

## (c) The effects of status on cooperation dynamics

We found support for our first prediction: status increased the odds of being nominated as a cooperation partner (OR = 1.06, CI = (1.00, 1.13)). Status also increased the odds of naming others as cooperation partners (OR = 1.12, CI = (1.02, 1.22)). There was no meaningful tendency towards selection-based homophily according to status; thus, status similarity was not associated with the odds of cooperation partnerships forming. Overall, our results suggest that higher status associates with both nominating and being nominated as a cooperation partner.

Cross-sectional analysis of Tsimane men's cooperation network in another community, using exponential random graph modelling (ERGM), is generally consistent with our longitudinal results (electronic supplementary material, table S8). Status again increased the odds of being nominated as a cooperation partner (OR = 1.05, CI = (1.01, 1.09)). However, status in this second community decreased the odds of nominating others as cooperation partners (OR = 0.95, CI = (0.92, 0.99)). See the electronic supplementary material for details.

## (d) The effects of cooperation on status dynamics

As shown in table 1, there was a substantial amount of change in men's status over time, with a change in status in period 1 being relatively similar to change in status in period 2. The insignificant shape effects indicate that status was not self-reinforcing. If anything, there is a weak tendency for status to regress to the mean. In addition to the shape effects, we find support for our second prediction. The average alter effect indicated that the average status of the individuals with whom an actor cooperates increased the odds of an individual's status ascending (OR = 1.20, CI = (1.01, 1.43)). In other words, our results indicate that an individual's status tended to rise to become similar to that of their cooperation partners through a process of network influence. The average alter effect is not directed so it accounts for both in-degree and out-degree ties. The effect was relatively unchanged if we instead specified an average in alter effect (average status of individuals who nominate the focal actor as a cooperation partner), a total alter effect (the summed statuses of an individual's cooperation partners) or a total in alter effect (summed statuses of individuals who nominate the focal actor as a cooperation partner). See the electronic supplementary material for these additional models. Being physically stronger and larger

(OR = 1.19, CI = (1.01, 1.40)), income (OR = 1.06, CI = (1.02, 1.11)) and log age (OR = 2.46, CI = (1.01, 5.99)) were also independently associated with increased status over time.

## 3. Discussion

Using a novel social network approach, we provide, to our knowledge, the first longitudinal evidence in a small-scale human society that inter-individual differences in men's status associate with the formation of cooperation partnerships, and that changes in a man's status associate with the statuses of his cooperation partners. However, these status-related effects were not large, relative to other predictors of cooperation partnerships, including kinship and network reciprocity. The latter is the tendency for an individual to nominate a cooperation partner who had previously nominated him but is not necessarily evidence of a strategy of reciprocity on the part of individuals. The relative magnitudes of status, kinship and network reciprocity effects within our study bear a similarity to the results of cross-sectional network analyses of cooperation in other small-scale societies. Within each of these other studies [19,20,42–45], status or income had smaller effects on cooperation compared to kinship and metrics of reciprocity. The cross-cultural similarity in how much status matters compared to kinship and reciprocity suggest some commonality in the formation of cooperation networks, despite large differences in network density (less than 0.01–0.39) and average degree (3.4–13.1) across studies.

We also find a tendency for cooperation partners to form transitive groups. While the transitive nominations of cooperation in our study do not explicitly capture transitive flows of aid, our result is consistent with a preference by Tsimane men to extend cooperation to their cooperation partners' cooperation partners. Recent evidence from longitudinal social network analysis has highlighted that the importance of transitivity is equivalent to that of network reciprocity for the formation of cooperation networks, i.e. friendships [46]. Whether transitivity in social networks in part reflects a cooperation strategy independent of reciprocity requires further theoretical inquiry. The present findings highlight the need for future research to consider transitive group formation, in both empirical investigation and theoretical modelling of the evolution of cooperation.

Among the Tsimane, men who are respected and influential in community decision-making (i.e. high in status) were more likely to be nominated over time as sharing food or assisting in hunting, fishing or horticultural labour. This result replicated in a cross-sectional analysis of Tsimane men in a second community. One explanation of these results is that higher-status Tsimane men may be more likely to share food and seek out cooperation opportunities with diverse community members. Political influence may hinge on maintaining coalitional support via generosity [16] and widespread social networking. Previous work with the Tsimane highlighted that politically influential men have reputations for generosity, and the association between a reputation for generosity and political influence appeared to be largely mediated by men's coalitional support [17]. But why seek respect or political influence in this relatively egalitarian context? Immediate benefits include resolving conflicts or steering community debates in directions that, while favourable to the community, are particularly favourable to oneself or family members. Among the

Tsimane, these debates often concern interactions with outside groups and conflicts over arable land, sexual jealousy, adultery, theft and other conflicts. Longer-term benefits may include lower chronic stress [47] and reproductive gains within and outside marital unions [48], which may be owing to enhanced mate value and greater social support for one's family during periods of particular need, such as illness [14,15].

Higher-status individuals may also be desirable as cooperation partners. Higher status among Tsimane men (in the longitudinal but not cross-sectional analysis) was associated with reports of receiving cooperation from others, and we further find that having higher-status cooperation partners was associated with an increase in an individual's status over time. The gain in status from cooperation with higher-status group members does not appear to be driven by a man's cooperativeness in general: the average status of a man's cooperation partners had a similar effect as his partners' summed statuses. The acquisition of social status may thus at least partly be a function of social connectedness to high-status group members.

A number of theoretical possibilities are consistent with status increasing as a result of cooperation with higher-status individuals. Lower-status individuals may gain access to information [10], resources or coalitional support that may increase their own status. Also, if high-status individuals are foci of social attention within their community, cooperation with high-status individuals may more effectively broadcast prosociality or other desirable attributes to other community members. A previous study of adults in an industrialized society found that individuals are more generous with better-connected members of their social network [49]. Individuals who engage in cooperation with higher status others may also mimic their prosocial behaviours [31], which may increase similarity in prestige and thus in status.

The associations between status and cooperation are independent of the associations between cooperation and other attributes of Tsimane men, including their age, income and physical strength and size. In the longitudinal analysis, older age was associated with fewer nominations of cooperation partners, which may reflect the increasing concentration of cooperation among close neighbours as older men's productivity and mobility wane. Strength and size associate with political influence in the Tsimane [17], which may be owing to their contribution to leader charisma [50] and coordination ability [51]. Indeed, strength and size in the present analysis associated with increases in status over time. However, physical strength and size also related to the acquisition of cooperation partners independent of Tsimane men's status, perhaps because physical formidability is desirous in an ally in the event of conflict. Strength and size may also associate with production skill, as may also be the case for hunting ability in other small-scale societies [52].

Because we only assessed men, no conclusions can be made about cooperation among women or between the sexes. Previous research among the Tsimane suggests that networks of cooperation differ in their structure across the sexes [37,53], in part owing to a sexual division of labour. Thus, future work should not simply analyse both sexes within a single network but consider the interaction and association of individuals within and across networks, which may inform men's and women's cooperation decisions.

In general, the present findings suggest that models of the evolution of cooperation should consider (i) the networked structure of human cooperation as well as (ii) the relative status of cooperators. Our findings do not exclude other mechanisms by which status hierarchy and cooperation catalyse each other. The opportunity to gain prestige may incentivize individuals to absorb costs of leadership during collective action [6,7]. For example, prior work with the Tsimane suggests that leader–follower relationships may help resolve collective action problems, particularly when leaders have traits that lower the costs and increase the efficacy of coordinating and enforcing cooperation [25,51].

Our findings further highlight the importance of longitudinal design for understanding processes of homophily in networks [54]. While mathematical models have suggested that cooperation is sustained when individuals selectively assort with those who display similarity in cooperativeness [55], the methodological paradigms used in field studies and experiments often cannot parse effects of network selection and network influence on actor similarity. Social ties may influence cooperation independent of tie selection: cooperation can increase or decrease via social influence [56]. Indeed, a longitudinal study of Hadza foragers suggests that cooperators assort within camps, owing to a process that may be more similar to network influence than network selection [57,58]. By using SAOMs, our study provides an important methodological contribution to the extant literature, highlighting an analytical strategy that can parse network selection and network influence.

We speculate on the implications of the present findings for the evolution of human status hierarchy, particularly the political egalitarianism characteristic of many small-scale societies. A prominent explanation of human egalitarianism, supported by ethnographic observation [59] and analytical modelling [60], suggests that egalitarianism emerged as humans evolved the cognitive abilities to form coalitions of sufficient size, efficacy and duration to check would-be dominants. In addition, egalitarianism may have been enabled by the evolution of greater interdependence among members of human groups, particularly in food production, group defence and raising of offspring [11]. Individuals who can supply valuable information or services in these contexts may receive deference [10] and be preferred as cooperation partners and mates [12], whereas pursuing dominance over others may risk losing cooperation partners essential to survival and reproduction. Furthermore, the transfer of information and resources from higher to lower-status individuals, as well as the potential reputational benefits to cooperating with higher-status individuals, may constrain or even erode status differentials. Analytic modelling suggests that status inequality is constrained when by cooperating, status-dissimilar individuals influence each other's statuses [61]. This process may counter the otherwise self-reinforcing nature of status (i.e. the 'Matthew effect'), whereby access to contested resources begets more status or is used to prevent others from acquiring status. In our longitudinal study in a Tsimane community, we find that individuals gain status the greater the status of their cooperation partners. This network influence effect is relatively independent of our estimation of general change in status in the community, which if anything shows a slight regression to the mean status over time. These results are consistent with the relative political egalitarianism we observe in Tsimane communities and with network influence playing a role in such egalitarianism. However, determining whether cooperation between status-dissimilar individuals is actually constraining increases in status inequality will require

**Figure 2.** The cooperation network over time, restricted to men present in at least two time waves. (a) 2009 (n = 60; ave. degree = 7.80; density = 0.087); (b) 2014 (n = 74; ave. degree = 3.39; density = 0.044); (c) 2017 (n = 70; ave. degree = 6.02; density = 0.072). Node size indicates the number of in-degree nominations an individual received for sharing food or assisting in hunting, fishing or horticultural labour. Node colour indicates the individual's status, such that darker colours reflect higher status. Arrows indicate the directionality of ties, i.e. incoming arrows indicate receipt of cooperation nominations. The digraphs were made in R package igraph [67]. The network at time wave 2 shows reduced density, which may correspond with the catastrophic flooding and crop loss that occurred that year. Men may have further concentrated cooperation with relatives, owing to mobility constraints or insufficient food to share widely. (Online version in colour.)

a more fine-grained study of how status change is distributed across the hierarchy.

Within and across human societies, socio-ecological variation in the sexual division of labour, subsistence strategies, mobility, community size and density, access to material wealth and other factors should pattern cooperation, status hierarchy and their dynamic interaction. For example, greater status inequality in societies with more material wealth may result in part from reduced cooperation between the wealthy and non-wealthy [62], which decreases the opportunity for lower-status individuals to gain status via network influence. Analytical models indicate that restricting connectivity in social networks can increase hierarchy [63,64]. Also, access to material wealth can make cooperation more overtly competitive or self-aggrandizing. In chiefdoms and Big Man societies, status often depended on sponsoring lavish feasts or gift exchanges, not only to signal personal qualities but also to generate indebtedness and reveal others' weaknesses [65]. The present study of the Tsimane provides a fruitful platform for future longitudinal studies in other societies, to further determine how particular cultural and ecological factors contribute to the relationship between cooperation and social status, and to infer the evolution of that relationship over human history.

# 4. Material and methods

## (a) Ethnographic setting

The Tsimane live in small villages in the neotropics of central, lowland Bolivia. Their economy is based on swidden horticulture (plantains, manioc, rice and corn), hunting, fishing and fruit gathering. Food sharing and collaboration in productive activities tend to be concentrated within extended families residing in the same or nearby households [66]. Women do the large majority of direct childcare and food processing. The Tsimane have no documented history of inter-village warfare. Within villages, conflicts tend to be resolved by the parties directly involved. For many of the conflicts that remain unresolved, third parties within the extended family or in the village may step in to help mediate. Villagers also hold meetings to respond to incursion by illegal loggers or other colonists, negotiate with itinerant merchants or coordinate projects with the Bolivian government or non-governmental organizations.

The Tsimane remained largely unconnected to Bolivian society until the mid-twentieth century, when a new wave of missionaries and a road from the highlands arrived. Average income is less than $2 (US) per day from the sale of horticultural products and sporadic wage labour with loggers and ranchers.

## (b) Data collection

Three waves of panel data, including status rankings and reported cooperation partners, were collected from the entire adult male population 21 years of age and above in one Tsimane village. Data were collected in 2009 (n = 72), 2014 (n = 78) and 2017 (n = 89). Growth in adult male population size was owing to immigration, in addition to more boys entering adulthood than adult men dying. A single wave of the panel data was collected in 2008 from the adult men in another Tsimane village (n = 89).

## (c) Cooperation networks

Networks of cooperation were constructed using a 'name generator' approach, with men freely listing other men within the village who had shared food with them or assisted them in hunting, fishing or horticultural labour over the previous six months. Men were questioned about each domain of cooperation separately, and we treated a nomination in any domain as a tie to the nominated individual. Networks were sociocentric, binary and directed. While Tsimane men's cooperation in hunting and fishing often produces shared rewards, we maintain nominations in these domains as directed because hunting and fishing partners may differ in their valuation of each other's cooperation, or in the caloric gain from cooperation [51]. For the longitudinal analysis, the population was restricted to adult men (ave. n = 68) who were present within the village in at least two waves of data collection (figure 2a–c). The composition change observed in the networks was modelled through the method of joiners and leavers [68].

## (d) Social status

At each wave of data collection, approximately a third of the adult men in the village were randomly selected to rank photos of other men from their village. Photos were Polaroids™ of the top-half of each man's body, set against as neutral a background as possible. Each ranker was presented two arrays of photos, one array at a time, and asked to rank the men in each array from highest to lowest (no ties allowed) according to 'who is most respected'. Each ranker also ranked two additional arrays of photos according

to 'whose voice carries the most weight during community debates.' Photos were numbered, and the photos chosen for a particular array corresponded to the row vectors in a matrix based on a projective plane (see the electronic supplementary material for more detail). Based on the matrix, each man's photo was ranked nine times, each time in an array with eight other photos to whom he had not yet been compared. Thus, each man could receive a status score ranging from 9 (lowest) to 81 (highest). This procedure kept the number of arrays and photos per array to a minimum, while ensuring every photo was compared to every other photo only once. To fit the current requirements of the modelling approach, the status rankings were transformed into ordinal percentile rankings with 10 levels. Because of the larger number of men, we used a larger matrix in 2014 and 2017, whereby each man's photo was ranked 10 times, in arrays with nine other photos. The photo-ranked scores from 2014 and 2017 were transformed to match the potential range in scores (9–81) from 2009 and in village 2.

Our confidence in the validity of the photo-ranking is strengthened by ethnographic observation. In the village assessed longitudinally, influence ranking in 2009 predicted who spoke more frequently during community meetings attended by C.R.v.R. that year ($r = 0.53$, $p < 0.001$, $n = 73$). The same was true for influence ranking and community meetings attended by C.R.v.R. in 2014 ($r = 0.67$, $p = 0.002$, $n = 49$).

## (e) Covariates

All demographic data used to determine the age of individuals and construct their kinship networks come from reproductive history interviews first collected from 2003 to 2005 and updated annually thereafter. Individuals were analysed as close kin if they were brothers or father and son (whether consanguineal or in-law). This definition of kinship reflects the concentration of Tsimane social life within clusters of households consisting of siblings, their spouses, their parents and their children. At each wave, household income was determined by asking men about all potential sources of income over the past year, including wage labour and sales of forest and horticultural goods. Income was transformed into ordinal percentile rankings with 10 levels. Every 1–3 years, clinicians employed with the Tsimane Health and Life History Project (THLHP: http://www.unm.edu/~tsimane) measured participants' height and weight with a portable stadiometer and a digital weigh scale, respectively. Shoulder and chest strength were measured with a Lafayette Manual Muscle Tester and grip strength was measured with a Smedley III dynamometer. We summed these values to create a composite upper body strength measure. A maximum-likelihood factor analysis indicated that height, weight and upper body strength comprised a distinct factor with adequate internal consistency. We standardized and averaged these measures to assess them as a single covariate that captures physical strength and size. Our models use the log of men's age as previous research has found that age does not have a linear association with status [17] or with food production [66]. See the electronic supplementary material for covariate descriptives and details of the factor analysis.

## (f) Analytical strategy

The longitudinal analysis was conducted using SAOMs [41] using RSIENA software (Simulation Investigation for Empirical Network Analysis: RSIENA version 1.2-12 in R 3.5.2) [69]. SAOMs estimate latent changes in a network, and changes in actors' 'behaviours' within that network, between observed time points. These changes are modelled as a continuous-time process, and actors are assumed to control their outgoing ties and the expression of 'behaviours' in a succession of multiple small steps, termed 'microsteps' [41]. These microsteps provide some statistical power for these complex models [70]. SAOMs estimate the rate of opportunities for change and further estimate the node-level (individual actor), dyad-level and network-level mechanisms that may be driving change using an evaluation function. A positive parameter estimate would indicate that the parameter produced a preference for creation and maintenance of a tie or an increase in a tie-dependent 'behaviour'. For the present analysis, a network-behaviour coevolutionary SAOM was specified [71], in order to simultaneously assess the temporal relationships that cooperation network dynamics (the dependent network) have with status (the dependent 'behaviour'). Several endogenous, structural network tendencies and several theoretically relevant actor-attribute effects—specifically those relating to kinship, log age, income and physical strength and size—were controlled for within the evaluation function. The group mean of the actor covariates is subtracted from each actor's score; these centred scores allow for more comparable estimates between covariates [69]. The model had good convergence, with an overall maximum convergence $t$-ratio of 0.13 and individual parameter $t$-ratios less than 0.10, and the model controlled for some time-heterogeneity of effects [72]. See the electronic supplementary material for a brief model description and further details on model specification.

For the cross-sectional network analysis, we used an ERGM. See the electronic supplementary material for details.

Ethics. Fieldwork was approved by the institutional review boards of the University of California, Santa Barbara and the University of Richmond, and by the Tsimane governing council. All study participants provided informed consent, and methods were approved at village meetings.

Data accessibility. Data are available at https://github.com/danielRedhead/dynamics-cooperation-status-analysis.

Authors' contributions. C.R.v.R. generated the theory, collected the data and drafted the manuscript. D.R. generated the theory, analysed the data and drafted the manuscript. R.O.G. edited the manuscript. H.K. collected data. M.G. edited the manuscript and collected the data.

Competing interests. We declare we have no competing interests.

Funding. Research was supported by grants from the National Science Foundation (NSF) (BCS-0136274, BCS-0422690, DDIG-0921429) and the National Institutes of Health (NIH/NIA) (R01AG024119-01).

Acknowledgements. We thank the residents of our study communities in Bolivia for their generous participation in this research. We also thank Joseph Kilgallen, Sarah Alami, Dr Paul Hooper and members of the Tsimane Health and Life History Project who helped collect data. Dr Ben Purzycki, Dr David Nolin, Dr Tom Pollett, Dr Gerulf Rieger, Natalia Federova, Riana Minocher, Jeffrey Andrews, Han Tran and four anonymous reviewers provided helpful comments on an earlier draft.

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
