## [Reviewer comments · Proceedings of the Royal Society B: Biological Sciences]

Review History

RSPB-2019-0330.R0 (Original submission)

Review form: Reviewer 1

Recommendation

Accept with minor revision (please list in comments)

Scientific importance: Is the manuscript an original and important contribution to its field?

Good

General interest: Is the paper of sufficient general interest?

Good

Quality of the paper: Is the overall quality of the paper suitable?

Excellent

Is the length of the paper justified?

Yes

Should the paper be seen by a specialist statistical reviewer?

Yes

Do you have any concerns about statistical analyses in this paper? If so, please specify them explicitly in your report.

No

It is a condition of publication that authors make their supporting data, code and materials available - either as supplementary material or hosted in an external repository. Please rate, if applicable, the supporting data on the following criteria.

Is it accessible?

No

Is it clear?

N/A

Is it adequate?

N/A

Do you have any ethical concerns with this paper?

No

Comments to the Author

The authors present an 8-year longitudinal analysis of male cooperative networks amongst Tsimane foragers. They analyze the data with a special focus on the interaction between status and cooperation.

I would like to applaud the authors on a very exciting and rigorously executed project. The topic is, in my view highly relevant and the data and results are an important contribution to the scientific community.

Introduction:

The introduction is clear. Aims, theoretical background, and hypotheses are clearly stated.

Design and statistics:

The study setup is well designed. The use combination of interviews and photo-ranked status measures is convincing. I should state, that I am not an expert in the statistical methods used (SAOMs, ERGMs). Hence, my interpretation of results and discussion is based on the authors' presentation. An expert in these methods should be consulted.

Discussion

The authors present a level-headed discussion, arguing for an effect of inter-individual variation in. status on cooperation networks, without neglecting large effects of reciprocity and kinship as well as effects of transitivity. In lines 191 following, the authors present explanations of why high-status men among the Tsimane are more often chosen as cooperative partners. I recommend treading more cautiously here, to not make too strong a claim about the direct causal relationship between individual status an inclusion in cooperation. Accepting my limited understanding of the statistics applied, I still wonder about some of the mediator variables that were included, such as physical strength, which impacts both status and inclusion in cooperation, as well as other possible mediating factors such as skill or personality. Of course, not all factors can be included, but a cautious discussion should be achieved. The authors mention alternatives along those lines

briefly in lines 227 and 239. Hence, all information is there, but not always present in the way the causation between status and cooperation is discussed.

All in all, this is a formidable manuscript.

Review form: Reviewer 2

Recommendation

Major revision is needed (please make suggestions in comments)

Scientific importance: Is the manuscript an original and important contribution to its field?

Good

General interest: Is the paper of sufficient general interest?

Acceptable

Quality of the paper: Is the overall quality of the paper suitable?

Good

Is the length of the paper justified?

Yes

Should the paper be seen by a specialist statistical reviewer?

Yes

Do you have any concerns about statistical analyses in this paper? If so, please specify them explicitly in your report.

Yes

It is a condition of publication that authors make their supporting data, code and materials available - either as supplementary material or hosted in an external repository. Please rate, if applicable, the supporting data on the following criteria.

Is it accessible?

N/A

Is it clear?

N/A

Is it adequate?

N/A

Do you have any ethical concerns with this paper?

No

Comments to the Author

This paper presents valuable data on the relationship between social status and cooperation, and is worth publishing. However, the paper is posed as testing predictions from theory, but the theory is vague and poorly specified, and to the extent to which I could figure it out, is not based on fitness maximization. The authors assume that people will assign higher status to cooperators

who produce group benefits, even in public goods contexts. No explanation is give for this assumption. Why should selection favor such a psychology? It does not categorize the status hierarchies of other primates. Alpha baboons are not alpha because they provide group benefits--it's because they can dominate other individuals, and the behavior of dominants can be quite deleterious to average fitness, for example in the case of infanticides. I agree that it seems to be the case that in sedentary horticultural societies high status individuals are generous, but this does not follow from any fitness maximizing model that I know of. The only mechanistic theory that he authors cite is the cultural model of Henrich et al in which this generosity arises as a side effect of an otherwise adaptive cultural learning mechanism. The authors need to rewrite and either provide real theory, or be clearer about its lac

Review form: Reviewer 3

Recommendation

Major revision is needed (please make suggestions in comments)

Scientific importance: Is the manuscript an original and important contribution to its field?

Acceptable

General interest: Is the paper of sufficient general interest?

Acceptable

Quality of the paper: Is the overall quality of the paper suitable?

Acceptable

Is the length of the paper justified?

Yes

Should the paper be seen by a specialist statistical reviewer?

No

Do you have any concerns about statistical analyses in this paper? If so, please specify them explicitly in your report.

No

It is a condition of publication that authors make their supporting data, code and materials available - either as supplementary material or hosted in an external repository. Please rate, if applicable, the supporting data on the following criteria.

Is it accessible?

No

Is it clear?

N/A

Is it adequate?

N/A

Do you have any ethical concerns with this paper?

No

Comments to the Author

This study investigates social network correlations over time between helping behaviors and prestige among men in an Amazonian preindustrial forager-horticulturalist society during three separate years 2009, 2014, and 2017. This is a fantastic dataset. The authors show interesting evidence that men in this society can raise their social status by forming cooperative relationships with men of higher relative social status. While it has long been appreciated that helping others can raise one's status, this finding suggests that the status of the recipient also matters. The authors suggest that status "diffuses" through cooperative relationships, but I was not completely convinced of this analogy.

The argument was hard to follow at several places, so I have several suggestions for improving the clarity of the writing and interpretation of the results. Most importantly, I do not understand why the authors constructed the cooperation network the way they did. The most straightforward way to make a cooperation network is to define the directed edge A to B as the amount or probability of help from A to B.

Since the authors collected data on multiple kinds of help (e.g. food sharing, help with labor), they could have made a network for each form of helping, allowing them to test whether different forms of help have different effects. Instead they treated any form of help as help (presumably because there were highly correlated??). Next, they defined the edge from A to B based on "nominations", where a nomination means that A reported that B previously helped A. In other words, the actor is the recipient of the help. This makes the results quite confusing.

Defining the network this way might also violate the assumptions of their model because actors in a stochastic actor-oriented model are assumed to control their outgoing edges, but individuals are not in control of who helped them. This might be based on a misunderstanding of their analysis, but in any case, it is so much clearer to just test helping, rather than testing acts of nominations of receiving help from specific helpers. This framing of the network makes the results section difficult to parse and also longer and more convoluted than necessary. Consider the simplest example: rather than simply saying "individuals were likely to help close kin" the authors' analysis compels them to write "individuals were likely to nominate close kin as cooperation partners" and as the effects become more complex, the writing becomes quite convoluted. For example:

"inter-individual differences in the number of times one is nominated as a cooperater (indegree) and the number of cooperators one nominates (outdegree) were not self-reinforcing"

I do understand that "indegree" (of nomination) actually means "outdegree" (of helping) but beyond that, I don't know what is being said here. It could mean several different things.

There is also a bit of possible confusion from using the same phrase to mean different things. To most readers of this journal, the phrase "evolution of cooperation" means genetic evolution of cooperative traits (or behaviors), but the authors repeatedly use this phrase to mean "emergence" or "change" in the cooperation network over time. There is a discussion of reciprocity and transitivity which conflates two similar but different meanings of reciprocity (lines 187 – 190). The authors state that "transitivity remains under-theorized relative to reciprocity within the evolution of cooperation" but this is because the term "reciprocity" refers to both a correlational property of networks (the symmetry of directed edges) and to a strategy of conditional costly helping based on receiving help in return (which is not directly observed in network data without experiment).

"Reciprocity" in the network sense can result from many kinds of mechanisms including helping

based on kinship, proximity, or any symmetrical variable. It is the strategy of reciprocity (i.e. reciprocal altruism) which has important theoretical basis in evolution of cooperation.

The network definition of reciprocity, like transitivity, can be trivial or interesting depending on what actually causes the pattern. For example, if a network is based on proximity, then both transitivity and reciprocity must occur as a byproduct. If helping is driven by spatial or genetic distance, then if A helps B due to proximity, and B helps C due to proximity, then A must also logically be somewhat close to C. Unlike reciprocity, there is no equivalent causal transitivity strategy in social evolution theory.

I did not understand the argument that status is transmitting or diffusing from high-status to low-status, rather than just increasing in the low-status individuals as we would expect. Is the status in high-status individuals decreasing? If so, could this just be regression to the mean? If not, then what is the diffusion/transmission argument based on?

I would suggest emphasizing that this study is only about men in the abstract or title.

I would also suggest being more conservative in using causal language to describe observational results (even if they are changes over time). For example, I do not agree that this study actually "shows" that "cooperation between individuals is motivated in part by opportunities to acquire status" (line 24) even if the results are consistent with that hypothesis.

Does Table 1 show all the effects that were tested or only the detected ("significant") effects? It is better to report the number of total number of p-values that were estimated.

I think the discussion would benefit from interpreting the results also in terms of theories about individual-level strategies (e.g. reputation-based partner choice, biological market theory, costly signalling) rather than only discussing emergent mechanisms at the group level.

Finally, I would suggest that the authors describe the effect sizes to help interpret the results using the odds ratios or converting them to probabilities (and/or confidence intervals). For example, if the odds ratio is 1 to 9 (and $p=0.052$), then rather than saying "there was a marginally significant tendency for those perceived high in status to be nominated as cooperators" ($p=0.052$), it would be better to say something like "we estimated that high status men were 0-10% more likely to help (95% CI of odds ratio= -0.01 to 0.11, $p=0.052$)". This will help the reader understand the actual importance of the effect.

I could not access the raw data or analysis.

Minor comments by line

Abstract

Line 23. Wording in first part is a bit awkward and also I do not think the data directly show these things. For example, it is an interpretation to say that status "transmits" through cooperative partnerships, rather than saying that it simply increases in the helper. Use of term 'co-evolved' is unclear.

Main text

33 first sentence is a bit awkward

38 opportunities?

62 replace "evolution" with different word like "emergence"

63 same

87 Did men report helping or status of others first? It seems plausible to me that this could have

some effect.

102 Reword for clarity. I think the author's compared more than the distribution of status?

106 Reword for clarity and explain terms

Supplement

In the ERGM, how many missing data were imputed? What proportion were missing? What were the results if these observations were simply removed?

Review form: Reviewer 4

Recommendation

Major revision is needed (please make suggestions in comments)

Scientific importance: Is the manuscript an original and important contribution to its field?

Excellent

General interest: Is the paper of sufficient general interest?

Good

Quality of the paper: Is the overall quality of the paper suitable?

Acceptable

Is the length of the paper justified?

Yes

Should the paper be seen by a specialist statistical reviewer?

No

Do you have any concerns about statistical analyses in this paper? If so, please specify them explicitly in your report.

No

It is a condition of publication that authors make their supporting data, code and materials available - either as supplementary material or hosted in an external repository. Please rate, if applicable, the supporting data on the following criteria.

Is it accessible?

No

Is it clear?

No

Is it adequate?

No

Do you have any ethical concerns with this paper?

No

Comments to the Author

In this paper, the authors investigate the hypothesis that cooperation and social status are co-emergent properties of human social groups, in that they are, to some degree, reciprocally causal.

In other words, individuals can gain cooperation opportunities by virtue of their social status, and gain in social status through cooperation. The authors investigate three implications of this hypothesis (1) social status influences choice of cooperation partners (2) cooperation is motivated in part by a desire to acquire higher social status, and (3) cooperation partners will come to have more similar social statuses by virtue of their cooperative act. To test these predictions, the authors applied stochastic actor-oriented models (SAOM) to three waves of panel data on the same Tsimane social group. They supported this dynamic analysis with a static analysis using ERGMs applied to a separate Tsimane social group. They found that many factors influenced choice of cooperation partners, including income, age, kinship, and physical strength/size. Additionally, and most relevant to the hypothesis being tested, individuals showed a marginally significant trend to preferentially nominate cooperation partners of high status, and the summed social statuses of individuals nominating a focal actor as a cooperation partner predicted increased social status for the focal actor. The authors conclude that their results support the hypothesis, and suggest that the diffusion of social status from high-status to lower-status individuals via cooperation leads to more egalitarian societies.

Overall, I find this to be a well written and thoughtfully-conducted study. As someone who studies social behavior in non-human animals, I don't feel qualified to evaluate the procedures by which the data were collected, although to my eye the data collection procedures appear sound. I do, however, have some thoughts about ways to improve the paper regarding the analysis and interpretation of results. I have divided my feedback into Major and Minor Points.

Major Points

The authors claim to have shown that "relative status affects with whom individuals cooperate," but the results do not support this claim as strongly they appear to suggest. Firstly, the relevant parameter estimate was only marginally significant; this is not damning, but should be more candidly stated in the abstract and discussion. The marginal nature of this result is stated only once in the Results, and otherwise this relationship is discussed as though the results provided unqualified support for the authors' hypothesis (e.g., lines 191-193).

On a related note, the outdegree effect of status on cooperation networks has a larger parameter estimate than the marginally significant indegree effect, yet is under-interpreted. Why do we see a difference in the effect of indegree and outdegree? Because the underlying cooperative behavior is in actuality a non-directed behavior (if A truly went fishing with B, B must have gone fishing with A), I would expect these effects to be very similar. Is the disparity between indegree and outdegree attributable to biased/erroneous reporting of cooperative activity, or could something else be driving this difference? The discussion focuses on high-status individuals being preferentially targeted for cooperation (status indegree), but very little time is spent interpreting the result that high-status individuals are more likely to nominate cooperators (status outdegree).

Similarly, turning to the portion of the paper modeling the effect of cooperation on status dynamics, why was total alter (i.e., the summed status of individuals identifying the focal individual as a cooperator) included in the model but not the inverse (i.e., the summed status of individuals that focal individual identified as a cooperator)? The authors should report this parameter estimate as well, even if it must be included in a secondary model due to collinearity.

Finally, I think some additional analyses of the effect of cooperation on status dynamics are in order. The current measure ("Status total alter") could be affected by the number of cooperators (in/outdegree) or by the value of cooperators. In the discussion, the authors talk about diffusion of social status via cooperative ties, but is it not possible that cooperation improves social status, regardless of the value of the cooperator? Perhaps, rather than social status diffusing from high-status to low-status individuals, engaging in cooperation per se increases status. Additionally, the

conclusion that “status transmits through cooperative partnerships, resulting in similarity between connected individuals over time” (abstract) is somewhat vague as to the nature of this transmittance, and has implications that the authors may not be meaning to imply. Do high-status individuals lose status when it is “transmitted” or it “diffuses” to low-status cooperation partners? If two low-status individuals cooperate together, does their status not change? I think the nature of this relationships could be partially clarified by running the SAOM with other cooperation-related covariates (e.g., mean status of cooperators, number of cooperators, mean status relative to status of focal individual). Finally, I think the discussion on diffusion should be changed to rely less heavily on metaphor; although useful, this metaphor has implications that may differ from reader to reader. Not to beat a dead horse here, but to me ‘diffusion’ implies a finite amount of status that flows from one individual to the other, causing a reduction in status by one individual and a gain by the other.

Minor Points

The second point in the abstract “cooperation between individuals is motivated in part by opportunities to acquire status”: I’m not sure where this is demonstrated in the paper. The distinction between point (i) and (ii) in the abstract appears to come down to (i) refers to whether cooperation is influenced by status and (ii) refers to whether individuals are *motivated* to cooperate by access to status. To my eye, the data reflect cooperative behavior and not motivation. I think the consideration of motivation underlying cooperative actions is best saved for the Discussion.

There are unexplained discrepancies in the sample sizes listed throughout the paper:

73 men in 2009 (line 334) vs 72 (line 299) vs 60 (Figure 2 caption)
 [60 (2009), 74 (2014), 70 (2017); Fig 2, Table 1] vs [global mean = 80; line 85] vs [72 (2009), 78 (2014), 2017 (89); line 299] vs. [global mean = 68; line 311]

It is possible that some of these discrepancies are explained by the text “who were present within the village in at least two waves of data collection” (lines 311-312) but this could be explained closer to the top of the manuscript. If this is true for all data presented, maybe the authors can only present the reduced sample size, as the discrepancy between the summary numbers in the table and the global mean reported on line 85 jumped out as incongruous to me.

ln. 178-181 “...the observed similarity in effect sizes of kinship and reciprocity relative to status and income...”: it’s unclear to me what point the authors are trying to make here. It doesn’t follow that the similarity in effect sizes of these parameters implies “commonality in the drivers of cooperation across disparate cultures and ecologies.” Are the authors comparing their observed parameters to those from other studies? Do they mean similarity among these parameter estimates? This should be clarified.

ln. 148: It would help to provide the corresponding parameter estimate from the reduced model. The fact that the parameter estimate is nearly unchanged in the reduced model (but confidence interval is reduced) strengthens the authors’ interpretation that collinearity is driving the wide confidence interval on the parameter estimate. Thus, I think reporting this would strengthen the paper.

ln. 194-196 “One explanation of these results...”: It is unclear to me if the authors are speculating with this sentence or reporting a previously reported finding. If the former, I would replace “are more likely” with “may be more likely” to clarify the speculative nature of this claim. If the latter, a citation is needed.

ln. 198 “whose effect on influence”: what word is “whose” modifying in this sentence? This could be clarified.

ln. 214-217 "In support of these arguments..": I don't see how this sentence supports the previous two. How does the fact that high-status individuals report more cooperators suggest that "cooperation with high status individuals may more effectively broadcast prosociality". This whole paragraph was difficult to follow and could be improved.

Decision letter (RSPB-2019-0330.R0)

26-Apr-2019

Dear Dr von Rueden:

I am writing to inform you that your manuscript RSPB-2019-0330 entitled "The dynamics of cooperation and social status in a small-scale society" has, in its current form, been rejected for publication in Proceedings B.

This action has been taken on the advice of referees, who have recommended that substantial revisions are necessary. Although the reviewers are uniformly enthusiastic about your manuscript, there are some substantive concerns regarding the underlying theory, the analysis, and your conclusions. With this in mind we would be happy to consider a resubmission, provided the comments of the referees are fully addressed. However please note that this is not a provisional acceptance.

Sincerely,

Prof Sarah F Brosnan
Editor, Proceedings B
mailto: proceedingsb@royalsociety.org

Associate Editor

Comments to Author:

Thank you for submitting your work to PRSB. I have now received comments on your MS from four experts in the field and have read your paper carefully myself. We all agree that your MS tackles an interesting research question and presents findings from a valuable data set. However, the reviewers raised a number of important concerns about many aspects of the paper. The reviews are clear and detailed so I will not repeat them here. However, I would like to draw your attention to four main categories of concern. The first, raised by R2, is that your study lacks a strong theoretical foundation. The second, raised by R1 and R4, is that conclusions have been overstated and results should be interpreted with more caution. The third, raised by R3 & R4, is that there are problems with the current analyses and these may need to be reworked. And, finally, R3 and R4 highlight the need for more conceptual clarity throughout the MS. In particular, these reviewers point out confusion surrounding the "diffusion" claim, both raising questions about what, precisely, diffusion means in this context. Revising the MS in line with these comments will be a serious undertaking. However, if the authors believe these comments can be addressed, I would be open to receiving a substantially revised version of the paper.

Reviewer(s)' Comments to Author:

Referee: 1

Comments to the Author(s)

The authors present an 8-year longitudinal analysis of male cooperative networks amongst Tsimane foragers. They analyze the data with a special focus on the interaction between status and cooperation.

I would like to applaud the authors on a very exciting and rigorously executed project. The topic is, in my view highly relevant and the data and results are an important contribution to the scientific community.

Introduction:

The introduction is clear. Aims, theoretical background, and hypotheses are clearly stated.

Design and statistics:

The study setup is well designed. The use combination of interviews and photo-ranked status measures is convincing. I should state, that I am not an expert in the statistical methods used (SAOMs, ERGMs). Hence, my interpretation of results and discussion is based on the authors' presentation. An expert in these methods should be consulted.

Discussion

The authors present a level-headed discussion, arguing for an effect of inter-individual variation in status on cooperation networks, without neglecting large effects of reciprocity and kinship as well as effects of transitivity. In lines 191 following, the authors present explanations of why high-status men among the Tsimane are more often chosen as cooperative partners. I recommend treading more cautiously here, to not make too strong a claim about the direct causal relationship between individual status an inclusion in cooperation. Accepting my limited understanding of the statistics applied, I still wonder about some of the mediator variables that were included, such as physical strength, which impacts both status and inclusion in cooperation, as well as other possible mediating factors such as skill or personality. Of course, not all factors can be included, but a cautious discussion should be achieved. The authors mention alternatives along those lines

briefly in lines 227 and 239. Hence, all information is there, but not always present in the way the causation between status and cooperation is discussed.

All in all, this is a formidable manuscript.

Referee: 2

Comments to the Author(s)

This paper presents valuable data on the relationship between social status and cooperation, and is worth publishing. However, the paper is posed as testing predictions from theory, but the theory is vague and poorly specified, and to the extent to which I could figure it out, is not based on fitness maximization. The authors assume that people will assign higher status to cooperators who produce group benefits, even in public goods contexts. No explanation is given for this assumption. Why should selection favor such a psychology? It does not categorize the status hierarchies of other primates. Alpha baboons are not alpha because they provide group benefits--it's because they can dominate other individuals, and the behavior of dominants can be quite deleterious to average fitness, for example in the case of infanticides. I agree that it seems to be the case that in sedentary horticultural societies high status individuals are generous, but this does not follow from any fitness maximizing model that I know of. The only mechanistic theory that the authors cite is the cultural model of Henrich et al in which this generosity arises as a side effect of an otherwise adaptive cultural learning mechanism. The authors need to rewrite and either provide real theory, or be clearer about its lack.

Referee: 3

Comments to the Author(s)

This study investigates social network correlations over time between helping behaviors and prestige among men in an Amazonian preindustrial forager-horticulturalist society during three separate years 2009, 2014, and 2017. This is a fantastic dataset. The authors show interesting evidence that men in this society can raise their social status by forming cooperative relationships with men of higher relative social status. While it has long been appreciated that helping others can raise one's status, this finding suggests that the status of the recipient also matters. The authors suggest that status "diffuses" through cooperative relationships, but I was not completely convinced of this analogy.

The argument was hard to follow at several places, so I have several suggestions for improving the clarity of the writing and interpretation of the results. Most importantly, I do not understand why the authors constructed the cooperation network the way they did. The most straightforward way to make a cooperation network is to define the directed edge A to B as the amount or probability of help from A to B.

Since the authors collected data on multiple kinds of help (e.g. food sharing, help with labor), they could have made a network for each form of helping, allowing them to test whether different forms of help have different effects. Instead they treated any form of help as help (presumably because there were highly correlated??). Next, they defined the edge from A to B based on "nominations", where a nomination means that A reported that B previously helped A. In other words, the actor is the recipient of the help. This makes the results quite confusing.

Defining the network this way might also violate the assumptions of their model because actors in a stochastic actor-oriented model are assumed to control their outgoing edges, but individuals are not in control of who helped them. This might be based on a misunderstanding of their

analysis, but in any case, it is so much clearer to just test helping, rather than testing acts of nominations of receiving help from specific helpers. This framing of the network makes the results section difficult to parse and also longer and more convoluted than necessary. Consider the simplest example: rather than simply saying “individuals were likely to help close kin” the authors’ analysis compels them to write “individuals were likely to nominate close kin as cooperation partners” and as the effects become more complex, the writing becomes quite convoluted. For example:

“inter-individual differences in the number of times one is nominated as a cooperator (indegree) and the number of cooperators one nominates (outdegree) were not self-reinforcing”

I do understand that “indegree” (of nomination) actually means “outdegree” (of helping) but beyond that, I don’t know what is being said here. It could mean several different things.

There is also a bit of possible confusion from using the same phrase to mean different things. To most readers of this journal, the phrase “evolution of cooperation” means genetic evolution of cooperative traits (or behaviors), but the authors repeatedly use this phrase to mean “emergence” or “change” in the cooperation network over time. There is a discussion of reciprocity and transitivity which conflates two similar but different meanings of reciprocity (lines 187–190). The authors state that “transitivity remains under-theorized relative to reciprocity within the evolution of cooperation” but this is because the term “reciprocity” refers to both a correlational property of networks (the symmetry of directed edges) and to a strategy of conditional costly helping based on receiving help in return (which is not directly observed in network data without experiment).

“Reciprocity” in the network sense can result from many kinds of mechanisms including helping based on kinship, proximity, or any symmetrical variable. It is the strategy of reciprocity (i.e. reciprocal altruism) which has important theoretical basis in evolution of cooperation.

The network definition of reciprocity, like transitivity, can be trivial or interesting depending on what actually causes the pattern. For example, if a network is based on proximity, then both transitivity and reciprocity must occur as a byproduct. If helping is driven by spatial or genetic distance, then if A helps B due to proximity, and B helps C due to proximity, then A must also logically be somewhat close to C. Unlike reciprocity, there is no equivalent causal transitivity strategy in social evolution theory.

I did not understand the argument that status is transmitting or diffusing from high-status to low-status, rather than just increasing in the low-status individuals as we would expect. Is the status in high-status individuals decreasing? If so, could this just be regression to the mean? If not, then what is the diffusion/transmission argument based on?

I would suggest emphasizing that this study is only about men in the abstract or title.

I would also suggest being more conservative in using causal language to describe observational results (even if they are changes over time). For example, I do not agree that this study actually “shows” that “cooperation between individuals is motivated in part by opportunities to acquire status” (line 24) even if the results are consistent with that hypothesis.

Does Table 1 show all the effects that were tested or only the detected (“significant”) effects? It is better to report the number of total number of p-values that were estimated.

I think the discussion would benefit from interpreting the results also in terms of theories about individual-level strategies (e.g. reputation-based partner choice, biological market theory, costly signalling) rather than only discussing emergent mechanisms at the group level.

Finally, I would suggest that the authors describe the effect sizes to help interpret the results using the odds ratios or converting them to probabilities (and/or confidence intervals). For example, if the odds ratio is 1 to 9 (and $p=0.052$), then rather than saying “there was a marginally significant tendency for those perceived high in status to be nominated as cooperators” ($p=0.052$), it would be better to say something like “we estimated that high status men were 0-10% more likely to help (95% CI of odds ratio= -0.01 to 0.11, $p=0.052$)”. This will help the reader understand the actual importance of the effect.

I could not access the raw data or analysis.

Minor comments by line

Abstract

Line 23. Wording in first part is a bit awkward and also I do not think the data directly show these things. For example, it is an interpretation to say that status “transmits” through cooperative partnerships, rather than saying that it simply increases in the helper. Use of term ‘co-evolved’ is unclear.

Main text

33 first sentence is a bit awkward

38 opportunities?

62 replace “evolution” with different word like “emergence”

63 same

87 Did men report helping or status of others first? It seems plausible to me that this could have some effect.

102 Reword for clarity. I think the author’s compared more than the distribution of status?

106 Reword for clarity and explain terms

Supplement

In the ERGM, how many missing data were imputed? What proportion were missing? What were the results if these observations were simply removed?

Referee: 4

Comments to the Author(s)

In this paper, the authors investigate the hypothesis that cooperation and social status are co-emergent properties of human social groups, in that they are, to some degree, reciprocally causal. In other words, individuals can gain cooperation opportunities by virtue of their social status, and gain in social status through cooperation. The authors investigate three implications of this hypothesis (1) social status influences choice of cooperation partners (2) cooperation is motivated in part by a desire to acquire higher social status, and (3) cooperation partners will come to have more similar social statuses by virtue of their cooperative act. To test these predictions, the authors applied stochastic actor-oriented models (SAOM) to three waves of panel data on the same Tsimane social group. They supported this dynamic analysis with a static analysis using ERGMs applied to a separate Tsimane social group. They found that many factors influenced choice of cooperation partners, including income, age, kinship, and physical strength/size. Additionally, and most relevant to the hypothesis being tested, individuals showed a marginally significant trend to preferentially nominate cooperation partners of high status, and the summed social statuses of individuals nominating a focal actor as a cooperation partner predicted increased social status for the focal actor. The authors conclude that their results support the

hypothesis, and suggest that the diffusion of social status from high-status to lower-status individuals via cooperation leads to more egalitarian societies.

Overall, I find this to be a well written and thoughtfully-conducted study. As someone who studies social behavior in non-human animals, I don't feel qualified to evaluate the procedures by which the data were collected, although to my eye the data collection procedures appear sound. I do, however, have some thoughts about ways to improve the paper regarding the analysis and interpretation of results. I have divided my feedback into Major and Minor Points.

Major Points

The authors claim to have shown that "relative status affects with whom individuals cooperate," but the results do not support this claim as strongly they appear to suggest. Firstly, the relevant parameter estimate was only marginally significant; this is not damning, but should be more candidly stated in the abstract and discussion. The marginal nature of this result is stated only once in the Results, and otherwise this relationship is discussed as though the results provided unqualified support for the authors' hypothesis (e.g., lines 191-193).

On a related note, the outdegree effect of status on cooperation networks has a larger parameter estimate than the marginally significant indegree effect, yet is under-interpreted. Why do we see a difference in the effect of indegree and outdegree? Because the underlying cooperative behavior is in actuality a non-directed behavior (if A truly went fishing with B, B must have gone fishing with A), I would expect these effects to be very similar. Is the disparity between indegree and outdegree attributable to biased/erroneous reporting of cooperative activity, or could something else be driving this difference? The discussion focuses on high-status individuals being preferentially targeted for cooperation (status indegree), but very little time is spent interpreting the result that high-status individuals are more likely to nominate cooperators (status outdegree).

Similarly, turning to the portion of the paper modeling the effect of cooperation on status dynamics, why was total alter (i.e., the summed status of individuals identifying the focal individual as a cooperator) included in the model but not the inverse (i.e., the summed status of individuals that focal individual identified as a cooperator)? The authors should report this parameter estimate as well, even if it must be included in a secondary model due to collinearity.

Finally, I think some additional analyses of the effect of cooperation on status dynamics are in order. The current measure ("Status total alter") could be affected by the number of cooperators (in/outdegree) or by the value of cooperators. In the discussion, the authors talk about diffusion of social status via cooperative ties, but is it not possible that cooperation improves social status, regardless of the value of the cooperator? Perhaps, rather than social status diffusing from high-status to low-status individuals, engaging in cooperation per se increases status. Additionally, the conclusion that "status transmits through cooperative partnerships, resulting in similarity between connected individuals over time" (abstract) is somewhat vague as to the nature of this transmittance, and has implications that the authors may not be meaning to imply. Do high-status individuals lose status when it is "transmitted" or it "diffuses" to low-status cooperation partners? If two low-status individuals cooperate together, does their status not change? I think the nature of this relationships could be partially clarified by running the SAOM with other cooperation-related covariates (e.g., mean status of cooperators, number of cooperators, mean status relative to status of focal individual). Finally, I think the discussion on diffusion should be changed to rely less heavily on metaphor; although useful, this metaphor has implications that may differ from reader to reader. Not to beat a dead horse here, but to me 'diffusion' implies a finite amount of status that flows from one individual to the other, causing a reduction in status by one individual and a gain by the other.

Minor Points

The second point in the abstract “cooperation between individuals is motivated in part by opportunities to acquire status”: I’m not sure where this is demonstrated in the paper. The distinction between point (i) and (ii) in the abstract appears to come down to (i) refers to whether cooperation is influenced by status and (ii) refers to whether individuals are *motivated* to cooperate by access to status. To my eye, the data reflect cooperative behavior and not motivation. I think the consideration of motivation underlying cooperative actions is best saved for the Discussion.

There are unexplained discrepancies in the sample sizes listed throughout the paper:

73 men in 2009 (line 334) vs 72 (line 299) vs 60 (Figure 2 caption)
 [60 (2009), 74 (2014), 70 (2017); Fig 2, Table 1] vs [global mean = 80; line 85] vs [72 (2009), 78 (2014), 2017 (89); line 299] vs. [global mean = 68; line 311]

It is possible that some of these discrepancies are explained by the text “who were present within the village in at least two waves of data collection” (lines 311-312) but this could be explained closer to the top of the manuscript. If this is true for all data presented, maybe the authors can only present the reduced sample size, as the discrepancy between the summary numbers in the table and the global mean reported on line 85 jumped out as incongruous to me.

ln. 178-181 “...the observed similarity in effect sizes of kinship and reciprocity relative to status and income...”: it’s unclear to me what point the authors are trying to make here. It doesn’t follow that the similarity in effect sizes of these parameters implies “commonality in the drivers of cooperation across disparate cultures and ecologies.” Are the authors comparing their observed parameters to those from other studies? Do they mean similarity among these parameter estimates? This should be clarified.

ln. 148: It would help to provide the corresponding parameter estimate from the reduced model. The fact that the parameter estimate is nearly unchanged in the reduced model (but confidence interval is reduced) strengthens the authors’ interpretation that collinearity is driving the wide confidence interval on the parameter estimate. Thus, I think reporting this would strengthen the paper.

ln. 194-196 “One explanation of these results...”: It is unclear to me if the authors are speculating with this sentence or reporting a previously reported finding. If the former, I would replace “are more likely” with “may be more likely” to clarify the speculative nature of this claim. If the latter, a citation is needed.

ln. 198 “whose effect on influence”: what word is “whose” modifying in this sentence? This could be clarified.

ln. 214-217 “In support of these arguments..”: I don’t see how this sentence supports the previous two. How does the fact that high-status individuals report more cooperators suggest that “cooperation with high status individuals may more effectively broadcast prosociality”. This whole paragraph was difficult to follow and could be improved.

Author's Response to Decision Letter for (RSPB-2019-0330.R0)

See Appendix A.

RSPB-2019-1367.R0

Review form: Reviewer 2

Recommendation

Accept as is

Scientific importance: Is the manuscript an original and important contribution to its field?

Excellent

General interest: Is the paper of sufficient general interest?

Good

Quality of the paper: Is the overall quality of the paper suitable?

Excellent

Is the length of the paper justified?

Yes

Should the paper be seen by a specialist statistical reviewer?

Yes

Do you have any concerns about statistical analyses in this paper? If so, please specify them explicitly in your report.

No

It is a condition of publication that authors make their supporting data, code and materials available - either as supplementary material or hosted in an external repository. Please rate, if applicable, the supporting data on the following criteria.

Is it accessible?

N/A

Is it clear?

N/A

Is it adequate?

N/A

Do you have any ethical concerns with this paper?

No

Comments to the Author

This is an excellent paper on an important topic. It should definitely be published.

Review form: Reviewer 4

Recommendation

Accept as is

Scientific importance: Is the manuscript an original and important contribution to its field?

Excellent

General interest: Is the paper of sufficient general interest?

Excellent

Quality of the paper: Is the overall quality of the paper suitable?

Excellent

Is the length of the paper justified?

Yes

Should the paper be seen by a specialist statistical reviewer?

No

Do you have any concerns about statistical analyses in this paper? If so, please specify them explicitly in your report.

No

It is a condition of publication that authors make their supporting data, code and materials available - either as supplementary material or hosted in an external repository. Please rate, if applicable, the supporting data on the following criteria.

Is it accessible?

Yes

Is it clear?

Yes

Is it adequate?

Yes

Do you have any ethical concerns with this paper?

No

Comments to the Author

The authors have done a thoughtful job of responding to my prior comments. I feel that this article is now a strong contribution to Proc B and to the broader literature. I have just a one small comment:

l. 58-59 The Mathew effect isn't about 'evaluation' per se, so I would restate this sentence such that it is more ambiguous about the mechanism of the hypothesized self-reinforcing effects.

Decision letter (RSPB-2019-1367.R0)

08-Jul-2019

Dear Dr von Rueden

I am pleased to inform you that your manuscript RSPB-2019-1367 entitled "The dynamics of

men's cooperation and social status in a small-scale society" has been accepted for publication in Proceedings B.

The referees have recommended publication, but one also suggest a minor revision. Therefore, I invite you to respond to that referee's comment. Because the schedule for publication is very tight, it is a condition of publication that you submit the revised version of your manuscript within 7 days. If you do not think you will be able to meet this date please let us know.

In order to ensure effective and robust dissemination and appropriate credit to authors the dataset(s) used should be fully cited. To ensure archived data are available to readers, authors

should include a 'data accessibility' section immediately after the acknowledgements section. This should list the database and accession number for all data from the article that has been made publicly available, for instance:

[http://datadryad.org/submit?journalID=RSPB&manu=\(Document not available\)](http://datadryad.org/submit?journalID=RSPB&manu=(Document+not+available)) which will take you to your unique entry in the Dryad repository. If you have already submitted your data to dryad you can make any necessary revisions to your dataset by following the above link. Please see <https://royalsociety.org/journals/ethics-policies/data-sharing-mining/> for more details.

Sincerely,

Dr Sarah Brosnan
Editor, Proceedings B
mailto: proceedingsb@royalsociety.org

Associate Editor
Comments to Author:

Thank you for submitting your revised MS. I sent your revision to two of the expert referees who reviewed your original submission. As you'll see from their comments, both referees feel that your revision addressed their comments and that the paper is now stronger as a result of your changes. Please note that R2 suggested one additional, minor change. The referees agree, as do I, that this paper would make an excellent contribution to Proc B.

Reviewer(s)' Comments to Author:

Referee: 2

Comments to the Author(s).

This is an excellent paper on an important topic. It should definitely be published.

Referee: 4

Comments to the Author(s).

The authors have done a thoughtful job of responding to my prior comments. I feel that this article is now a strong contribution to Proc B and to the broader literature. I have just a one small comment:

l. 58-59 The Mathew effect isn't about 'evaluation' per se, so I would restate this sentence such that it is more ambiguous about the mechanism of the hypothesized self-reinforcing effects.

Decision letter (RSPB-2019-1367.R1)

12-Jul-2019

Dear Dr von Rueden

I am pleased to inform you that your manuscript entitled "The dynamics of men's cooperation and social status in a small-scale society" has been accepted for publication in Proceedings B.

Open Access

Paper charges

Sincerely,

Appendix A

June 11, 2019

Christopher R. von Rueden

Associate Professor
Jepson School of Leadership Studies
University of Richmond

Daniel J. Redhead

Postdoctoral Fellow
Department of Human Behavior, Ecology and Culture
Max Planck Institute for Evolutionary Anthropology

Dear Dr. Brosnan and *Proc. R. Soc. B* Editorial Board,

We submit for your consideration a revision of manuscript RSPB-2019-0330, now entitled “The dynamics of men’s cooperation and social status in a small-scale society”.

We are happy that all four reviewers found the manuscript a valuable contribution, and we are grateful for the thoughtfulness of their comments, which prompted numerous changes to our manuscript (see our detailed responses to referees below). In general, we amended our paper to be more cautious in our conclusions and rewrote multiple sections for clarity or to provide greater explanation. We also analyzed several new models, which substantiated our initial conclusions. We also expand on some of the theoretical motivation. We believe these changes constitute significant revision which have improved the manuscript, and we hope you agree!

Our manuscript is one of the first longitudinal assessments of cooperation in a preindustrial human society. More specifically, we test (1) how the cooperation network of a forager-horticulturalist community changed over an eight-year period, as a function of the social status of cooperators and other theoretically-motivated processes including network reciprocity, transitivity, and kin favouritism. Simultaneously, we test (2) how the social status of individuals changed as a result of position within the cooperation network. Our use of stochastic actor oriented models (SAOMs) is uniquely suited to such tests. SAOMs allow us to parse the mechanisms by which assortment arises over time in a real-world network, which has been a goal of cooperation researchers for some time. Thus, we are able to show that cooperators are not assorting on the basis of status but rather influence each other’s statuses over time.

We are also able to show a reciprocal influence between status hierarchy and cooperation that heretofore has not been demonstrated. The upshot of this is that cooperation researchers can’t ignore hierarchy, and hierarchy researchers can’t ignore cooperation networks. We also discuss the implications of our results for the origins of egalitarianism in human societies, suggesting that status-driven cooperation can restrain the self-reinforcing tendency of hierarchy. The existing literature emphasizes a quite different mechanism (coordinated status leveling).

Our results will appeal to a broad swath of readership, particularly researchers in evolutionary ecology and anthropology who study the evolution of cooperation or hierarchy. We look forward to hearing from you regarding our resubmission.

Sincerely,

Christopher von Rueden and Daniel Redhead (joint first authors)

MS Reference Number: RSPB-2019-0330

Our responses to the Associate Editor and to the reviewers are bolded and in Times New Roman font. Following our responses, we have appended a copy of the original submission noting where changes were made.

Associate Editor

Comments to Author:

Thank you for submitting your work to PRSB. I have now received comments on your MS from four experts in the field and have read your paper carefully myself. We all agree that your MS tackles an interesting research question and presents findings from a valuable data set. However, the reviewers raised a number of important concerns about many aspects of the paper. The reviews are clear and detailed so I will not repeat them here. However, I would like to draw your attention to four main categories of concern. The first, raised by R2, is that your study lacks a strong theoretical foundation. The second, raised by R1 and R4, is that conclusions have been overstated and results should be interpreted with more caution. The third, raised by R3 & R4, is that there are problems with the current analyses and these may need to be reworked. And, finally, R3 and R4 highlight the need for more conceptual clarity throughout the MS. In particular, these reviewers point out confusion surrounding the “diffusion” claim, both raising questions about what, precisely, diffusion means in this context. Revising the MS in line with these comments will be a serious undertaking. However, if the authors believe these comments can be addressed, I would be open to receiving a substantially revised version of the paper.

We are glad to see that all four reviewers found the paper a valuable contribution. Their reviews were quite helpful. We’ve responded in detail below to each of the reviewers’ concerns or suggestions for improvement. We have amended our paper to be more cautious in our conclusions (concern two) and more clear with our definitions and explanations (concern four). We also analyzed several new models prompted by reviewer comments, which substantiated our initial conclusions (concern three). The results we present in Table 1 changed slightly as a result of us re-running our principal model with an average status alter effect, as opposed to a total status alter effect (though we include the latter in the SM, along with other models we tested). We also expand on some of the theoretical motivation (concern one). We believe these changes constitute significant revision which have improved the manuscript, and we hope you agree!

Reviewer(s)' Comments to Author:

Referee: 1

Comments to the Author(s)

The authors present an 8-year longitudinal analysis of male cooperative networks amongst Tsimane foragers. They analyze the data with a special focus on the interaction between status and cooperation.

I would like to applaud the authors on a very exciting and rigorously executed project.

The topic is, in my view highly relevant and the data and results are an important contribution to the scientific community.

Thank you!

Introduction:

The introduction is clear. Aims, theoretical background, and hypotheses are clearly stated.

Design and statistics:

The study setup is well designed. The use combination of interviews and photo-ranked status measures is convincing. I should state, that I am not an expert in the statistical methods used (SAOMs, ERGMs). Hence, my interpretation of results and discussion is based on the authors' presentation. An expert in these methods should be consulted.

Discussion

The authors present a level-headed discussion, arguing for an effect of inter-individual variation in status on cooperation networks, without neglecting large effects of reciprocity and kinship as well as effects of transitivity. In lines 191 following, the authors present explanations of why high-status men among the Tsimane are more often chosen as cooperative partners. I recommend treading more cautiously here, to not make too strong a claim about the direct causal relationship between individual status and inclusion in cooperation. Accepting my limited understanding of the statistics applied, I still wonder about some of the mediator variables that were included, such as physical strength, which impacts both status and inclusion in cooperation, as well as other possible mediating factors such as skill or personality. Of course, not all factors can be included, but a cautious discussion should be achieved. The authors mention alternatives along those lines briefly in lines 227 and 239. Hence, all information is there, but not always present in the way the causation between status and cooperation is discussed.

We agree with your concerns about treading more cautiously regarding interpretation of our results. We no longer include as one of our principal findings the claim that men are motivated to cooperate as a means of increasing their status, since we did not directly test men's motivations. In the introduction, we presented several (evolutionary) reasons why we would expect status and cooperation to associate, and in the discussion we return to these explanations in light of our findings. Though we now take even more care to not argue beyond what our data analyses justify. We did not test for any mediation effects per se. Rather, we included other structural, dyadic, and individual-level variables in our model to determine the unique effects of status beyond these other covariates. Thus, we can discount

confounding by age, reciprocity, kinship, body size, income, etc., as well as describe the effect status has relative to these other covariates.

All in all, this is a formidable manuscript.

Thank you!

Referee: 2

Comments to the Author(s)

This paper presents valuable data on the relationship between social status and cooperation, and is worth publishing.

Thank you!

However, the paper is posed as testing predictions from theory, but the theory is vague and poorly specified, and to the extent to which I could figure it out, is not based on fitness maximization. The authors assume that people will assign higher status to cooperators who produce group benefits, even in public goods contexts. No explanation is given for this assumption. Why should selection favor such a psychology? It does not categorize the status hierarchies of other primates. Alpha baboons are not alpha because they provide group benefits---it's because they can dominate other individuals, and the behavior of dominants can be quite deleterious to average fitness, for example in the case of infanticides. I agree that it seems to be the case that in sedentary horticultural societies high status individuals are generous, but this does not follow from any fitness maximizing model that I know of. The only mechanistic theory that the authors cite is the cultural model of Henrich et al in which this generosity arises as a side effect of an otherwise adaptive cultural learning mechanism. The authors need to rewrite and either provide real theory, or be clearer about its lack.

In our Introduction, we mention several evolutionary (fitness maximizing) models that specify causal links from cooperation to social status, including costly signaling and partner-choice models. We then describe how other evolutionary (fitness maximizing) models predict the reverse causality, from status to cooperation. These include two models by Henrich et al.: one where cooperation by high status individuals is imitated, and another where high status individuals are preferential cooperation partners because individuals hope to acquire status from learning from these high status individuals. Our longitudinal study using SAOM analytic techniques is particularly useful in shedding light on one aspect of the latter model: does cooperation with high status individuals elevate one's own status? We present the first evidence in support of this.

Because our study was not designed to test the specific predictions of the various foregoing models against each other, we generalize our predictions to test commonalities to these models: 1) higher status individuals will more frequently be nominated as cooperation partners, and 2) cooperators gain status, particularly by cooperating with higher status

individuals. The longitudinal design and incorporation of multiple potentially confounding variables (whether structural or individual-level), increase our confidence in our results.

We discuss (in the opening paragraph) why the foregoing models won't generalize well to non-human hierarchy: humans are much more interdependent in things like skill acquisition and food production. Thus, individuals known for an ability and willingness to provide benefits to others are more likely to gain and maintain status. Furthermore, our adeptness at cultural learning increases the ways in which we can benefit each other. And we mention that humans coalitional abilities means group members are better able to keep those who act dominantly in check. We now elaborate more on these points.

Referee: 3

Comments to the Author(s)

This study investigates social network correlations over time between helping behaviors and prestige among men in an Amazonian preindustrial forager-horticulturalist society during three separate years 2009, 2014, and 2017. This is a fantastic dataset. The authors show interesting evidence that men in this society can raise their social status by forming cooperative relationships with men of higher relative social status. While it has long been appreciated that helping others can raise one's status, this finding suggests that the status of the recipient also matters. The authors suggest that status "diffuses" through cooperative relationships, but I was not completely convinced of this analogy.

Thank you for your comments in support of the manuscript. And your criticisms have been helpful in improving it. The finding that the status of one's cooperation partners affects one's own status is novel, and we agree that we need to amend how we describe this result. We no longer use the "diffusion" analogy, and more strictly refer to it as evidence of network influence (which is the appropriate terminology in the network literature). How this influence happens could be due to several possibilities, some of which we outline in our discussion, including access to cooperators' information or coalition support or more effective broadcasting of one's cooperativeness.

The argument was hard to follow at several places, so I have several suggestions for improving the clarity of the writing and interpretation of the results. Most importantly, I do not understand why the authors constructed the cooperation network the way they did. The most straightforward way to make a cooperation network is to define the directed edge A to B as the amount or probability of help from A to B.

We understand that our construction of the cooperation network differs from some approaches in the field of cooperation research, particularly approaches where networks are constructed from direct measurement of transfers of food or other aid between

individuals. However, our approach is common in the SIENA literature, e.g. studies assessing networks of advice, friendship, bullying, etc. For example, individuals are nominating a bully, but the actual direction of the bullying behaviour is the converse (e.g. Huitsing et al. 2014 “Victims bullies and their defenders” in Development and Psychopathology). In our study, we measured cooperation as the actor’s perception of their cooperative relationships, with the underlying events that comprise the relationship (the actual food transfers/aid in production) being latent/unobserved. This was methodologically more tractable, given the longitudinal depth of our study. More importantly, we wanted to measure cooperative relationships on the order of years rather than day-to-day quantities of aid given back and forth.

Since the authors collected data on multiple kinds of help (e.g. food sharing, help with labor), they could have made a network for each form of helping, allowing them to test whether different forms of help have different effects. Instead they treated any form of help as help (presumably because there were highly correlated??).

Yes, there was quite substantial overlap in these networks. Regardless, we were interested in looking at the totality of male-male cooperation in the community within a single analysis. It was not our goal in the current study to determine how different forms of cooperation associate with status over time, but rather how cooperation in general associates with status over time. Future work may investigate whether different networks have different effects.

Next, they defined the edge from A to B based on “nominations”, where a nomination means that A reported that B previously helped A. In other words, the actor is the recipient of the help. This makes the results quite confusing. Defining the network this way might also violate the assumptions of their model because actors in a stochastic actor-oriented model are assumed to control their outgoing edges, but individuals are not in control of who helped them. This might be based on a misunderstanding of their analysis, but in any case, it is so much clearer to just test helping, rather than testing acts of nominations of receiving help from specific helpers. This framing of the network makes the results section difficult to parse and also longer and more convoluted than necessary. Consider the simplest example: rather than simply saying “individuals were likely to help close kin” the authors’ analysis compels them to write “individuals were likely to nominate close kin as cooperation partners” and as the effects become more complex, the writing becomes quite convoluted. For example:

“inter-individual differences in the number of times one is nominated as a cooperator (indegree) and the number of cooperators one nominates (outdegree) were not self-reinforcing”

I do understand that “indegree” (of nomination) actually means “outdegree” (of helping) but beyond that, I don’t know what is being said here. It could mean several different things.

We agree that there are tradeoffs in analyzing and describing our results as we did. But this approach was appropriate to our method (peer nomination). Given the way we measured the networks, we're asking something equivalent to "who is your friend?" This actually is in keeping with the assumption that actors are controlling outgoing ties. If we analyzed the transpose of our network, as suggested, this would prevent actors from being able to control their ties. Alternatively, had we directly observed actual food transfers/aid in production and transformed them into binary directed networks, then we agree that with our approach the assumption would have been broken. However, we measured cooperation as the actor's perception of their cooperative relationships, with the underlying events that comprise the relationship (the actual food transfers/aid in production) being latent/unobserved. A large proportion of SIENA studies use a similar approach.

There is also a bit of possible confusion from using the same phrase to mean different things. To most readers of this journal, the phrase "evolution of cooperation" means genetic evolution of cooperative traits (or behaviors), but the authors repeatedly use this phrase to mean "emergence" or "change" in the cooperation network over time. There is a discussion of reciprocity and transitivity which conflates two similar but different meanings of reciprocity (lines 187—190). The authors state that "transitivity remains under-theorized relative to reciprocity within the evolution of cooperation" but this is because the term "reciprocity" refers to both a correlational property of networks (the symmetry of directed edges) and to a strategy of conditional costly helping based on receiving help in return (which is not directly observed in network data without experiment).

"Reciprocity" in the network sense can result from many kinds of mechanisms including helping based on kinship, proximity, or any symmetrical variable. It is the strategy of reciprocity (i.e. reciprocal altruism) which has important theoretical basis in evolution of cooperation.

The network definition of reciprocity, like transitivity, can be trivial or interesting depending on what actually causes the pattern. For example, if a network is based on proximity, then both transitivity and reciprocity must occur as a byproduct. If helping is driven by spatial or genetic distance, then if A helps B due to proximity, and B helps C due to proximity, then A must also logically be somewhat close to C. Unlike reciprocity, there is no equivalent causal transitivity strategy in social evolution theory.

We agree with these critiques of our terminology. We now only reference evolution when we mean to refer to genetic evolution of (cooperative) traits. And we more clearly interpret our reciprocity effect as "network reciprocity" rather than a strategy of reciprocity. The former may result from the latter, in part, but we abstain from any strong conclusions in this regard. While there is no causal transitivity strategy in social evolution theory, as you note, our finding of transitivity unfolding over time suggests there may be more at play than just geographic or genetic distance. But here too we refrain from any strong conclusions.

I did not understand the argument that status is transmitting or diffusing from high-status to low-status, rather than just increasing in the low-status individuals as we would expect. Is the status in high-status

individuals decreasing? If so, could this just be regression to the mean? If not, then what is the diffusion/transmission argument based on?

We no longer refer to this effect as “diffusion” but more strictly as a network influence effect. For whatever reason, men gain status by cooperating with higher status individuals. This doesn’t necessarily mean higher status men lose status when cooperating with lower status individuals. But since our status measures are zero-sum, this is a possibility. It depends on how much the status gain of lower status men cooperating with higher status men is offset by entry of young men or immigrants of low status into the adult network over time. We still speculate that this network influence effect can aid in maintaining an egalitarian social structure, but with more qualification.

I would suggest emphasizing that this study is only about men in the abstract or title.

Agreed and done!

I would also suggest being more conservative in using causal language to describe observational results (even if they are changes over time). For example, I do not agree that this study actually “shows” that “cooperation between individuals is motivated in part by opportunities to acquire status” (line 24) even if the results are consistent with that hypothesis.

Agreed. We have removed such causal language, from the abstract and elsewhere in the paper.

Does Table 1 show all the effects that were tested or only the detected (“significant”) effects? It is better to report the number of total number of p-values that were estimated.

Yes Table 1 shows all the effects that were tested.

I think the discussion would benefit from interpreting the results also in terms of theories about individual-level strategies (e.g. reputation-based partner choice, biological market theory, costly signalling) rather than only discussing emergent mechanisms at the group level.

Agreed. We mention these theories in the introduction when motivating our predictions, and now return to them in the discussion as well.

Finally, I would suggest that the authors describe the effect sizes to help interpret the results using the odds ratios or converting them to probabilities (and/or confidence intervals). For example, if the odds ratio is 1 to 9 (and $p=0.052$), then rather than saying “there was a marginally significant tendency for those perceived high in status to be nominated as cooperators ” ($p=0.052$), it would be better to say something like “we estimated that high status men were 0-10% more likely to help (95% CI of odds ratio= -0.01 to 0.11, $p=0.052$)”. This will help the reader understand the actual importance of the effect.

As suggested, we now use the ORs and CI in the main text.

I could not access the raw data or analysis.

We apologize for this. The data and analysis scripts should be available at <https://github.com/danielRedhead/dynamics-cooperation-status-analysis>.

Minor comments by line

Abstract

Line 23. Wording in first part is a bit awkward and also I do not think the data directly show these things. For example, it is an interpretation to say that status “transmits” through cooperative partnerships, rather than saying that it simply increases in the helper. Use of term ‘co-evolved’ is unclear.

Abstract amended as suggested.

Main text

33 first sentence is a bit awkward

38 opportunities?

62 replace “evolution” with different word like “emergence”

63 same

87 Did men report helping or status of others first? It seems plausible to me that this could have some effect.

102 Reword for clarity. I think the author’s compared more than the distribution of status?

106 Reword for clarity and explain terms

We also addressed these line specific suggestions. Regarding potential effects of the cooperation interviews on status-rankings, we describe evidence of external validity for the latter in the methods.

Supplement

In the ERGM, how many missing data were imputed? What proportion were missing? What were the results if these observations were simply removed?

We re-ran the ERGM without the imputed data (omitting participants with NAs) and results were qualitatively similar. We added this new analysis to the SM. We also state in SM the proportion of data missing: 18% for status, 7% for strength and size, and 1% for income. The missing status data in this second community is largely for men who infrequently interact with other members of the community, and whose residence in the community is sporadic.

Referee: 4

Comments to the Author(s)

In this paper, the authors investigate the hypothesis that cooperation and social status are co-emergent properties of human social groups, in that they are, to some degree, reciprocally causal. In other words, individuals can gain cooperation opportunities by virtue of their social status, and gain in social status through cooperation. The authors investigate three implications of this hypothesis (1) social status influences choice of cooperation partners (2) cooperation is motivated in part by a desire to acquire higher social status, and (3) cooperation partners will come to have more similar social statuses by virtue of their cooperative act. To test these predictions, the authors applied stochastic actor-oriented models (SAOM) to three waves of panel data on the same Tsimane social group. They supported this dynamic analysis with a static analysis using ERGMs applied to a separate Tsimane social group. They found that many factors

influenced choice of cooperation partners, including income, age, kinship, and physical strength/size. Additionally, and most relevant to the hypothesis being tested, individuals showed a marginally significant trend to preferentially nominate cooperation partners of high status, and the summed social statuses of individuals nominating a focal actor as a cooperation partner predicted increased social status for the focal actor. The authors conclude that their results support the hypothesis, and suggest that the diffusion of social status from high-status to lower-status individuals via cooperation leads to more egalitarian societies.

Overall, I find this to be a well written and thoughtfully-conducted study.

Thank you!

As someone who studies social behavior in non-human animals, I don't feel qualified to evaluate the procedures by which the data were collected, although to my eye the data collection procedures appear sound. I do, however, have some thoughts about ways to improve the paper regarding the analysis and interpretation of results. I have divided my feedback into Major and Minor Points.

Major Points

The authors claim to have shown that “relative status affects with whom individuals cooperate,” but the results do not support this claim as strongly they appear to suggest. Firstly, the relevant parameter estimate was only marginally significant; this is not damning, but should be more candidly stated in the abstract and discussion. The marginal nature of this result is stated only once in the Results, and otherwise this relationship is discussed as though the results provided unqualified support for the authors' hypothesis (e.g., lines 191-193).

We agree and have amended discussion of the results to better qualify the status-relevant effects, relative to other parameters in our model. We also now explicitly acknowledge the large effects of kinship, reciprocity, and transitivity in our abstract.

On a related note, the outdegree effect of status on cooperation networks has a larger parameter estimate than the marginally significant indegree effect, yet is under-interpreted. Why do we see a difference in the effect of indegree and outdegree? Because the underlying cooperative behavior is in actuality a non-directed behavior (if A truly went fishing with B, B must have gone fishing with A), I would expect these effects to be very similar. Is the disparity between indegree and outdegree attributable to biased/erroneous reporting of cooperative activity, or could something else be driving this difference?

We note in our methods that fishing and hunting, while collaborative activities, can plausibly involve unreciprocated nominations. This is in part because we're measuring people's memory of cooperation rather than actual flows of aid. It is also possible that one fishing partner places a higher value on the other's cooperation because of differential effort, or because the first partner gained a greater caloric return from the joint activity, which both may increase the likelihood of nomination or cross a threshold for what counts as providing cooperation in the minds of our participants. In the same way, friendship nominations are often not reciprocated in other network analyses from other societies (e.g. Almaatouq et al. 2016 “Are you your friends' friend?” in PLoS ONE). That said, the

difference between the indegree and outdegree effects are not large in our longitudinal sample.

The discussion focuses on high-status individuals being preferentially targeted for cooperation (status indegree), but very little time is spent interpreting the result that high-status individuals are more likely to nominate cooperators (status outdegree).

We dedicate a paragraph in the discussion to interpreting the status outdegree results, though it is only significant in the longitudinal sample. We argue that independent of their generosity, high status individuals are likely to be desirable as cooperation partners. And then we discuss some possibilities why this may be the case. One possibility is that one's own status rises as a result of targeting higher status individuals as cooperation partners. Indeed, this is what our analysis of status dynamics suggests.

Similarly, turning to the portion of the paper modeling the effect of cooperation on status dynamics, why was total alter (i.e., the summed status of individuals identifying the focal individual as a cooperator) included in the model but not the inverse (i.e., the summed status of individuals that focal individual identified as a cooperator)? The authors should report this parameter estimate as well, even if it must be included in a secondary model due to collinearity.

The total alter effect is not directed, so it reflects connection to any alters and how these alters' status impacts the focal actor's status. We clarify this in the text.

We now also analyze an in alter effect as you suggest. The relative equivalence of the alter and in alter effects suggests directionality of cooperation ties is not as important in terms of status dynamics.

Finally, I think some additional analyses of the effect of cooperation on status dynamics are in order. The current measure ("Status total alter") could be affected by the number of cooperators (in/outdegree) or by the value of cooperators. In the discussion, the authors talk about diffusion of social status via cooperative ties, but is it not possible that cooperation improves social status, regardless of the value of the cooperator? Perhaps, rather than social status diffusing from high-status to low-status individuals, engaging in cooperation per se increases status. Additionally, the conclusion that "status transmits through cooperative partnerships, resulting in similarity between connected individuals over time" (abstract) is somewhat vague as to the nature of this transmittance, and has implications that the authors may not be meaning to imply. Do high-status individuals lose status when it is "transmitted" or it "diffuses" to low-status cooperation partners? If two low-status individuals cooperate together, does their status not change? I think the nature of this relationships could be partially clarified by running the SAOM with other cooperation-related covariates (e.g., mean status of cooperators, number of cooperators, mean status relative to status of focal individual).

We now analyze both a total alter effect and an average alter effect. They are relatively equivalent, which suggests the amount of ties an individual has doesn't explain how the status of cooperative ties increases your status.

Finally, I think the discussion on diffusion should be changed to rely less heavily on metaphor; although useful, this metaphor has implications that may differ from reader to reader. Not to beat a dead horse here, but to me ‘diffusion’ implies a finite amount of status that flows from one individual to the other, causing a reduction in status by one individual and a gain by the other.

Agreed, we no longer refer to our network influence effect as diffusion, and we emphasize that the result doesn’t necessarily mean higher status individuals lose status when cooperating with lower status individuals.

Minor Points

The second point in the abstract “cooperation between individuals is motivated in part by opportunities to acquire status”: I’m not sure where this is demonstrated in the paper. The distinction between point (i) and (ii) in the abstract appears to come down to (i) refers to whether cooperation is influenced by status and (ii) refers to whether individuals are *motivated* to cooperate by access to status. To my eye, the data reflect cooperative behavior and not motivation. I think the consideration of motivation underlying cooperative actions is best saved for the Discussion.

We agree and have removed claims of demonstrating motivation in the abstract and elsewhere, though we speculate on motivation in the discussion.

There are unexplained discrepancies in the sample sizes listed throughout the paper:

73 men in 2009 (line 334) vs 72 (line 299) vs 60 (Figure 2 caption)
[60 (2009), 74 (2014), 70 (2017); Fig 2, Table 1] vs [global mean = 80; line 85] vs [72 (2009), 78 (2014), 2017 (89); line 299] vs. [global mean = 68; line 311]

It is possible that some of these discrepancies are explained by the text “who were present within the village in at least two waves of data collection” (lines 311-312) but this could be explained closer to the top of the manuscript. If this is true for all data presented, maybe the authors can only present the reduced sample size, as the discrepancy between the summary numbers in the table and the global mean reported on line 85 jumped out as incongruous to me.

Yes, the discrepancy is due to restriction of the analysis to men present in at least two waves. We now state this not only in the methods but also in the introduction to head off confusion.

In. 178-181 “...the observed similarity in effect sizes of kinship and reciprocity relative to status and income...”: it’s unclear to me what point the authors are trying to make here. It doesn’t follow that the similarity in effect sizes of these parameters implies “commonality in the drivers of cooperation across disparate cultures and ecologies.” Are the authors comparing their observed parameters to those from other studies? Do they mean similarity among these parameter estimates? This should be clarified.

We hope we have sufficiently clarified this section. Our point is that status appears to matter less as a predictor of cooperation ties relative to kinship or some metric of reciprocity, not only in our study but also in other network studies of cooperation in small-scale societies. One interpretation of this is that processes of social network formation have some uniformity cross-culturally even in the face of large socio-ecological differences.

In. 148: It would help to provide the corresponding parameter estimate from the reduced model. The fact that the parameter estimate is nearly unchanged in the reduced model (but confidence interval is reduced)

strengthens the authors' interpretation that collinearity is driving the wide confidence interval on the parameter estimate. Thus, I think reporting this would strengthen the paper.

Thanks- we now include the effect size from the reduced model in the main text in addition to SM.

ln. 194-196 "One explanation of these results...": It is unclear to me if the authors are speculating with this sentence or reporting a previously reported finding. If the former, I would replace "are more likely" with "may be more likely" to clarify the speculative nature of this claim. If the latter, a citation is needed.

Yes, this is speculation since we haven't directly measured men's motivations. We changed the wording accordingly.

ln. 198 "whose effect on influence": what word is "whose" modifying in this sentence? This could be clarified.

We rewrote this sentence for clarity.

ln. 214-217 "In support of these arguments..": I don't see how this sentence supports the previous two. How does the fact that high-status individuals report more cooperators suggest that "cooperation with high status individuals may more effectively broadcast prosociality". This whole paragraph was difficult to follow and could be improved.

We rewrote this section for clarity.